# Leaving no one behind: using action research to promote male involvement in maternal and child health in Iringa region, Tanzania

Stephen Maluka ,[1] Paul Japhet,[1] Sian Fitzgerald,[2] Khadija Begum,[3] May Alexander,[4] Peter Kamuzora[1]

► Prepublication history and additional materials for this paper is available online. To view these files, please visit the journal online (http://dx.doi.org/10.1136/bmjopen-2020-038823).

¹Institute of Development Studies, University of Dar es Salaam, Dar es Salaam, United Republic of Tanzania
²Healthbridge Foundation, Ottawa, Ontario, Canada
³Health Bridge Foundation, Ottawa, UK
⁴Iringa Region, Health Department, Iringa, United Republic of Tanzania

**Correspondence to**
Dr Stephen Maluka;
Stephenmaluka@yahoo.co.uk

## ABSTRACT

**Introduction** Male involvement has been reported to improve maternal and child health (MCH) outcomes. However, most studies in low-income and middle-income countries have reported low participation of men in MCH-related programmes. While there is a growing interest in the involvement of men in MCH, little is known on how male involvement can be effectively promoted in settings where entrenched unequal gender roles, norms and relations constrain women from effectively inviting men to participate in MCH.

**Methods and analysis** This paper reports participatory action research (PAR) aimed to promote male participation in pregnancy and childbirth in Iringa Region, Tanzania. As part of the Innovating for Maternal and Child Health in Africa project, PAR was conducted in 20 villages in two rural districts in Tanzania. Men and women were engaged separately to identify barriers to male involvement in antenatal care and during delivery; and then they were facilitated to design strategies to promote male participation in their communities. Along with the PAR intervention, researchers undertook a series of research activities. A thematic analysis was used to analyse the data. The common strategies designed were: engaging health facility committees; using male champions and male gatekeepers; and using female champions to sensitise and provide health education to women. These strategies were validated during stakeholders' meetings, which were convened in each community.

**Discussion** The use of participatory approach not only empowers communities to diagnose barriers to male involvement and develop culturally acceptable strategies but also increases sustainability of the interventions beyond the life span of the project. More lessons will be identified during the implementation of these strategies.

### Strengths and limitations of this study

► This study is one of the few to have used action research approach to promote male involvement in maternal and child health.
► The findings reported are based on the initial stages of the participatory action research.
► Another limitation of the study is lack of independence of the data collection because data collection was carried out by the researchers who were part of the action research team.

## INTRODUCTION

Maternal and child death is a global problem and is highly overwhelming in low-income and middle-income countries. An estimate of 303 000 pregnant women and 8.4 million under 5 children died in 2015.[1] Low-income and middle-income countries account for approximately 99% of maternal and child deaths.[2 3] In Tanzania, the maternal mortality rate is 556 per 100 000 live births while the infant and under-5 mortality rates are 43/1000 and 67/1000, respectively.[4] The neonatal mortality rate is 25 per 1000 live births.[4]

Low utilisation of maternal and child health (MCH) services is mentioned as one of the root causes of maternal and child deaths.[1–3] In Tanzania, for example, though over 90% of pregnant women attend antenatal care (ANC) at least once, only 51% complete four or more recommended visits. Additionally, only 24% of women attend their first ANC before the fourth month of pregnancy.[4 5] Health facility deliveries were 63% and 37% were home deliveries. Sixty-four per cent of deliveries were assisted by health professional and 36% were not.[4] Furthermore, only 13% of women who delivered at home attended postnatal care within the recommended timeline of 48 hours after delivery.[4]

In Tanzania, like in other African countries, the adverse effect of socioeconomic factors on maternal, newborn and child health is compounded by gender inequalities that limit women's autonomy and decision-making power.[4] This prevents women and their families from practising healthy behaviours as well as seeking and receiving life-saving care

BMJ

at and around the time of birth.[4 6–9] The delay in seeking care and delay in reaching healthcare facilities occurs when women lack support from male partners, especially when the situation involves the need for money.[9 10]

Male involvement has been reported to improve MCH outcomes. Among the perceived benefits associated with increased male involvement include increased access to ANC and delivery at the health facility with assistance of skilled birth attendants[11]; increased knowledge of both women and men on the obstetric care emergencies[12 13]; increased use of family planning (FP) methods[14 15] and decreased gender-related barriers to MCH services.[16 17]

Globally, a number of strategies have been implemented to increase male involvement in MCH services. The strategies include health education campaigns on the mass media, at work places, community outreach and counselling of couples at health facilities.[18 19] Others are fast tracking to couples who attend clinics together[18–22] and denying services to women attending ANC without male partners.[19–23]

In Tanzania, policies, guidelines and strategies emphasise male involvement in MCH services. The focused ANC guidelines require pregnant women to attend clinics with their partners as part of the prevention of mother to child transmission of HIV. Pregnant women are, therefore, required to attend the first ANC visits with their male partners for HIV testing.[24–26] Despite these strategies, only 30% of male partners participate in the couple counselling and HIV testing.[26]

While there is a growing interest in the involvement of men in MCH, little is known on how male involvement can be effectively promoted in settings where entrenched unequal gender roles, norms and relations constrain women from effectively inviting men to participate in MCH. Most policies take a narrow definition of male involvement focusing more on clinic attendance. In this study, male involvement refers to provision of social, informational and psychological support to pregnant women. This paper makes a valuable contribution to the literature on male involvement in MCH because it addresses the question of how male involvement can be promoted.

## Methods
### Study settings and design
This study was part of the large project under the Innovating for Maternal and Child Health in Africa (IMCHA) programme, which is being implemented in Kilolo and Mufindi districts in Iringa region, Tanzania. IMCHA programme (September 2015– March 2020) aimed to improve maternal, newborn and child health outcomes by strengthening health systems to become more equitable, using primary healthcare as an entry point. Iringa region was selected because of the earlier research collaboration between some of the IMCHA researchers and the regional and district decision-makers on strengthening decentralised district health management. The districts are predominantly rural and the majority of

**Table 1** Maternal and child health indicators in the study districts

| Indicator | National (DHS 2015/2016), % | Kilolo (HMIS), % | Mufindi (HMIS), % |
|---|---|---|---|
| ANC visits within 12 weeks | 24 | 27 | 16.8 |
| Four or more ANC visits | 51 | 27.1 | 23.6 |
| Facility delivery | 63 | 92.2 | 96.8 |
| Assisted by skilled birth attendants | 64 | 92 | 92 |
| Postnatal care within 48 hours | 31 | 76.2 | 51.8 |
| Male involvement in ANC HIV testing | 30 | Low | Low |
| Use of morden family planning | 32 | 30 | 30 |

ANC, antenatal care; DHS, Demographic and Health Survey; HMIS, Health Management Information System.

the population rely on agriculture for subsistence and production. Table 1 shows MCH indicators in the study districts.

A participatory action research (PAR) design was adopted in this study.[27] Action research is the research conducted in partnership with members of the community mainly aimed at bringing about structural or cultural change.[27 28] In PAR, researchers do not just conduct studies on community rather they form partnerships with community members to identify issues important to the local community, develop ways of studying them and take actions on the resulting knowledge.[28] The PAR process consisted of a series of meetings during which women, men and community stakeholders identified barriers to male involvement in MCH; and then designed strategies to promote male participation. The action research team (ART) led by the researchers from the University of Dar es Salaam, in collaboration with the regional and district health managers from Iringa Region facilitated the PAR. As indicated in figure 1, the PAR process involved five phases. As indicated in online supplemental file 1, in order to complete one cycle Women and men groups conducted eight meetings on a monthly basis.

### Introduction and formation of women and men groups
We conducted a series of consultation meetings with the community members in 20 intervention villages. Prior to our first community, we had several discussions within our team of how our social identities with respect to education, gender, sexual orientation and material resources would influence the PAR process. This was an important process as most of the research team members had not worked with communities. Before starting community meetings, we conduct previsits in all 20 intervention villages to discuss our research with community leaders. On entering the community, the ART first introduced the project to the community and facilitated the formation of women and men groups in 20 intervention villages. We used community leaders mainly village chairperson and

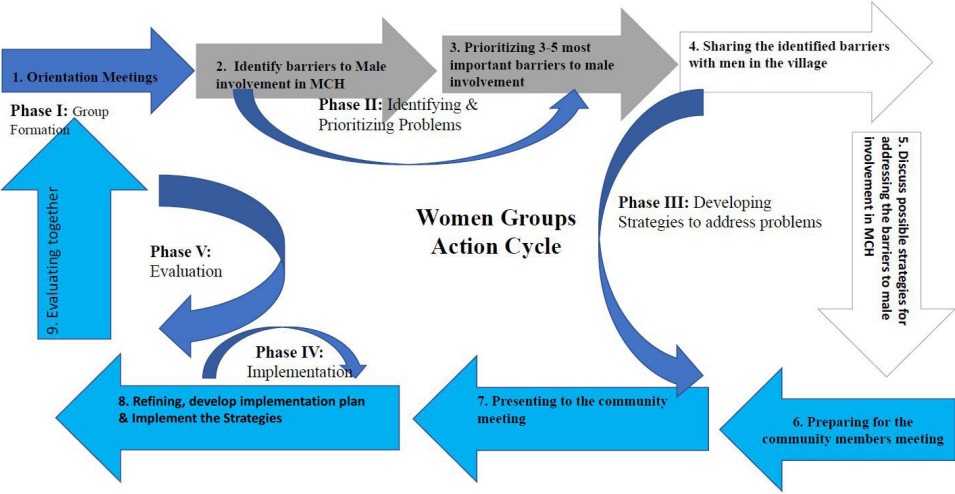

**Figure 1** Participatory action research cycle. MCH, maternal and child health.

village executive officers to persuade other community members to attend an initial meeting. In each village, women and men groups comprised 20 and 10 members, respectively, who were selected during community meetings facilitated by the ART. The criteria for selecting women included: age 15–49 years, experience in using MCH services, and women who were pregnant or had given birth in the last 12 months preceding the study. However, in a few villages women who were above 50 years were selected by community members. These were mostly traditional birth attendants and the community considered them important to be included in the women groups in order to share their experience. The criteria for selecting male participants included: having a wife/female partner and experience with childbirth. In total, 400 women and 200 men were identified in 20 intervention villages. These 400 women and 200 men made up several groups of 20 and 10 respectively. In addition, 40 community health workers (CHWs); 2 from each village were identified and included in the women groups. The selected women, men and CHWs were given training on how they could diagnose various barriers to the access and use of MCH services in their community; and then design strategies to address these barriers. The training was standardised and was provided by the experienced qualitative researchers (SM and PK) in collaboration with the district health managers.

### Diagnosing and prioritising problems
In phase II, the ART led the women groups in identifying and prioritising barriers to male involvement in MCH in their community. This phase involved two different meetings. In the first meeting, women were able to identify barriers without the presence of men. In the next meeting, the ART facilitated women groups to share and discuss the identified barriers with a group of 10 men (male champions) in their community. The rationale of including men in the meetings was to give them an opportunity to share their views as key stakeholders and decision makers in MCH matters. This process also aimed

to prepare men to actively participate in designing and implementation of the strategies.

### Designing male engagement strategies
In phase III, women and men together developed strategies to promote male involvement in MCH. In addition, the ART facilitated community meetings where women groups and male champions shared the barriers and strategies designed to promote male involvement. The key stakeholders who participated in the community meetings included the government and community leaders, religious leaders, health providers and members of the Health Facility Governing Committees. The objective of the community meetings was to give stakeholders an opportunity to discuss and refine the problems, root causes and strategies identified by women and men groups. In addition, the community meetings aimed at actively involving community stakeholders in the implementation of the strategies in their respective areas. As indicated in table 2, a total of 450 stakeholders from all 20 intervention villages participated in community meetings.

### Implementing and evaluating strategies
In phase IV, women and men groups developed plans and led the implementation of the strategies to promote male involvement in MCH. In phase V, women groups and male champions together evaluated the effectiveness of the strategies. In this paper, we report the first three phases of the PAR as the last two phases (implementation and evaluation) were not yet completed. As indicated in table 3, in total, we facilitated 102 meetings between February 2017 and March 2018 for the first three phases.

### Data collection techniques
As part of the PAR intervention, the researchers conducted a series of research activities. All the meetings conducted with women, men and community members were documented. In some cases, the discussions were audio recorded with permission from the participants. Where recording was not possible, notes from the

**Table 2** Participants by type attending community meetings

| Districts | Kilolo | Mufindi | Total |
|---|---|---|---|
| District representatives | 3 | 2 | 5 |
| Ward councillors | 5 | 4 | 9 |
| Ward executive officers | 5 | 5 | 10 |
| Community development officers | 3 | 1 | 4 |
| Village executive officers | 8 | 10 | 18 |
| Village chair persons | 10 | 9 | 19 |
| Religious leaders | 18 | 19 | 37 |
| Hamlet chairs (community leaders) | 52 | 49 | 101 |
| Primary school representatives | 6 | 11 | 17 |
| Secondary school representatives | 4 | 1 | 5 |
| Health facility staff (healthcare workers) | 5 | 4 | 9 |
| Health facility governing committees | 4 | 4 | 8 |
| Community health workers | 20 | 18 | 38 |
| Male champions (men's representatives) | 31 | 19 | 50 |
| Women groups representatives | 53 | 51 | 104 |
| Others | 11 | 5 | 16 |
| | 238 | 212 | 450 |

discussion were taken. In addition, trained research assistants prepared reports for the meetings. Similarly, all the presentations made by the women groups during community meetings were audio-recorded. Additionally, interviews and structured questionnaires were administered

**Table 3** Meetings facilitated by the ART in the communities

| | Type of meeting | No |
|---|---|---|
| 1 | Orientation and formation of women and men groups | 20 |
| 2 | Training of women group members to identify barriers and design strategies to address these barriers | 10 |
| 3 | Training of community health workers to supervise women and men in identifying barriers, designing strategies, implementation and evaluation of the strategies | 2 |
| 4 | Identification and prioritisation of barriers to MCH, including male involvement | 20 |
| 5 | Sharing identified barriers with men and jointly designing strategies | 20 |
| 6 | Preparing women group members for the community meetings (meeting logistics, preparing presentations, sociodramas, role plays, etc) | 20 |
| 7 | Community/stakeholders meetings to discuss and refine barriers, root causes and strategies | 10 |
| | Total meetings | 102 |

ART, action research team; MCH, maternal and child health.

to a subset of men and women group members. Furthermore, a structured questionnaire was used to collect data on sociodemographic characteristics, men's knowledge and awareness of FP and ANC services and their practices in relation to prenatal care and childbirth, and related decision-making power. Data were collected through digital tablets using Open Data Kit software by five local interviewers between June and July 2018. The same interviewers covered all 20 villages. Researchers got verbal consent from all the prospective respondents. Verbal consent was preferred because in these settings the use of written consent forms might be perceived by participants to be threatening.

### Data analysis

All recordings from the women, men and community meetings were transcribed verbatim by trained transcribers. The transcripts were reviewed by the core research team members and notes were made for each transcript. The transcripts and summaries of the stakeholders' meetings were analysed using a thematic approach.[29] First, the principal investigator (SM) developed a code manual based on the objectives of the study. The coding framework was further discussed and refined by the research team members. PJ and SM manually coded the transcripts and summaries of the meetings to the identified codes. Additional codes, which emerged during coding process, were added along the way. PK cross-checked the coding process. Responses were then compared across different categories of participants. Finally, data were summarised and synthesised, keeping the key expressions of respondents as illustrative cases. Quantitative data analyses were performed using StataV.15.0 (StataCorp).

### Patient and public involvement

This study was conducted in partnership with members of the community. The PAR process consisted of a series of meetings during which women, men and community stakeholders identified barriers to male involvement in MCH and then designed strategies to promote male participation. Findings were shared during stakeholders meetings held in the respective communities.

### FINDINGS

This section presents the key findings of the study. The key themes covered are: characteristics of women and men groups, barriers to male involvement in MCH and strategies designed to promote male involvement.

### Characteristics of the women group members

The mean age (±SD) of the women respondents was 35.7±9.4 (median=35, min=16 and max=61). Most of the women (75%) were between 26 and 49 years of age. About 78% (n=208) of the women were married, and most of them were living with their husbands (96%). Among the married women, 21% (n=44) reported that

| Table 4 | Characteristics of the women respondents | | |
|---|---|---|---|
| Characteristics | | N | % |
| Age category | | | |
| 16–25 years | | 41 | 15.5 |
| 26–35 years | | 98 | 37.0 |
| 36–49 years | | 100 | 37.7 |
| 50–65 years | | 26 | 9.8 |
| Marital status | | | |
| Never married | | 29 | 10.9 |
| Married | | 208 | 78.5 |
| Divorced/ separated | | 3 | 1.1 |
| Widows | | 25 | 9.4 |
| Highest level of education | | | |
| Never attended school | | 6 | 2.3 |
| Primary, not completed | | 15 | 5.7 |
| Primary, completed | | 191 | 72.1 |
| Secondary, not completed | | 22 | 8.3 |
| Secondary, completed | | 28 | 10.6 |
| Vocational/adult education | | 3 | 1.1 |

| Table 5 | Male demographic characteristics | | |
|---|---|---|---|
| Male demographic characteristics | | N | % |
| Age range | | | |
| 20–39 | | 90 | 51 |
| 40–59 | | 79 | 45 |
| Above 60 | | 6 | 3 |
| Level of education | | | |
| Not completed primary school | | 2 | 1 |
| Completed primary school | | 144 | 82 |
| Not completed secondary school | | 11 | 6 |
| Completed secondary school | | 15 | 9 |
| College/university | | 3 | 2 |
| Marital status | | | |
| Single | | 2 | 1 |
| Married | | 172 | 98 |
| Widower | | 1 | 1 |
| Occupation | | | |
| Employed | | 2 | 1 |
| Farmer | | 161 | 92 |
| Self-employed/ small business | | 12 | 7 |
| No of children | | | |
| 1–3 | | 85 | 49 |
| 4–6 | | 69 | 39 |
| 7+ | | 21 | 12 |

their husbands had more than one wife. Table 4 shows the main characteristics of the women group members.

In terms of ANC and childbirth, around 9% (23 of 265) of the women group members were pregnant during the interviews, and 19 (83%) of them reported to attend ANC. We also explored prenatal and childbirth care practices of these women during their immediate pregnancy. Although around 96% of the women (254 of 265) attended ANC at health facilities, only about 72% of them (184 of 254) had had four or more ANC visits and 93% (235 of 254) gave birth at health facilities. The prevalence of ≥4 ANC visits and facility births were low among all the women interviewed—69% and 91%, respectively.

### Characteristics of male champions

As indicated in table 5, about 51% (90) of the male champions were between the age range of 20 and 39. Moreover, 82% (144) had completed primary education and nearly all (98%) were married. As for women group members, the majority (92%) of the male champions relied on agriculture for subsistence and production.

We also assessed the awareness and knowledge of male champions on various MCH matters. As indicated in table 6, nearly all participants (99%) were aware of the FP. However, most men (65%) only mentioned condom as the main FP method. Other methods of FP planning were rarely mentioned. The majority of the participants (84%) understood the role of ANC as taking care of pregnant women and their fetus. However, only 21% (63 of 175) agreed that men were required to accompany their wives for ANC visits. Furthermore, 80% (125 of 156) had knowledge of exclusive breast feeding (EBF) and 78% (136 of 175) had knowledge of the duration of the EBF.

### Barriers to male involvement

While majority of women preferred to attend first ANC appointment with their partners, men did not wish to be actively involved in ANC and delivery. Participants perceived men as breadwinners and their main role in pregnancy and child birth was to support their partners financially. There were contradicting views regarding the need for men to assist their wives with household chores; while women preferred to be assisted with household chores during pregnancy and early days after birth, majority of male participants felt that this was mainly women's responsibility. Furthermore, it was evident that men did not prefer their wives to use modern FP methods due to fear of side effects. Men also had the perception that the use of FP would increase cheating for married women. Table 7 provides a summary of the barriers to male involvement as identified in most of the intervention villages.

### Strategies designed to promote male involvement

The key strategies which were commonly proposed were: engaging health facility governing committees; use of male champions and gatekeepers; and using women champions.

#### Engaging health facility governing committees

In all public primary healthcare facilities, there are health facility governing committees. The committees

**Table 6** Men's awareness and knowledge of maternal and child health matters

| Awareness and knowledge of male champions on MCH | N | % |
|---|---|---|
| Ever heard of family planning | | |
| Yes | 173 | 99 |
| No | 1 | 1 |
| Meaning of family planning | | |
| Control of family size | 62 | 26 |
| Child spacing | 145 | 61 |
| Preventing unwanted pregnancy | 32 | 13 |
| Men's role in family planning | | |
| Consent | 54 | 27 |
| Support | 145 | 73 |
| Others (specify) | 0 | 0 |
| I don't know | 0 | 0 |
| Awareness of contraceptive methods for men | | |
| Vasectomy | 10 | 5 |
| Male condom | 144 | 65 |
| Injectables | 32 | 14 |
| Diaphram | 4 | 2 |
| Hormonal contraception, such as the pill | 32 | 14 |
| What ANC entails | | |
| Taking care of pregnant women and their fetus | 162 | 84 |
| Giving drugs and injection to pregnant women | 9 | 5 |
| Detecting and managing complications | 21 | 11 |
| Men's role in ANC | | |
| Financial support | 24 | 8 |
| Encouraging and reminding pregnant women | 105 | 36 |
| Providing emotional and moral support | 103 | 35 |
| Accompanying pregnant women to clinic visits | 63 | 21 |
| Knowledge of exclusive breast feeding (EBF) | | |
| Breast milk alone | 125 | 80 |
| Breast milk with little water | 3 | 2 |
| No knowledge | 28 | 18 |
| Knowledge of the duration of EBF | | |
| Below 6 months | 13 | 7 |
| 6 months | 136 | 78 |
| Above 6 months | 10 | 6 |
| No knowledge | 16 | 9 |

ANC, antenatal care; MCH, maternal and child health.

are composed of representatives of the community and health facility. The health facility governing committees have the mandate of overseeing the delivery of quality healthcare services and mobilising the community to finance and manage healthcare services. The use of health facility governing committees was identified as a potential strategy to promote male engagement in MCH services. Participants reported that actively engaging health facility governing committees would help address health facility related barriers to male involvement including long waiting time, disrespectful languages of the health providers, and lack of physical space to accommodate men and women concurrently as exemplified by one of the stakeholders:

> The health governing committee should be coached on their roles and responsibilities to ensure they properly supervise provision of services at the health facility. If they are engaged, the committee will be able to talk to the health care workers not to harass pregnant women and their partners. The committee can also tackle other supply side related problems which hinder use of health care services. This will attract men to attend clinics (Male participant during stakeholder meeting).

### Engaging male champions and gatekeepers

There was consensus among participants that men do not accompany their wives to ANC clinics; and fear to use FP methods because they lack education. While women most often get health education in health facilities, particularly during ANC visits, men rarely receive health education. In addition, traditional gender roles and norms hinder men's participation in MCH matters. Men's perception of superiority makes it difficult for wives to pass the message from health providers to their husbands. Most often wives fear to discuss FP, pregnancy and child birth-related matters with their husbands due to culturally entrenched power differences between wives and husbands. In this context, male champions are better positioned to educate their fellow men and ultimately break the entrenched cultural gender roles and norms. Participants suggested that male champions should be identified in each village. They should also be facilitated in terms of key messages to be delivered to men. Male champions should deliver key messages through bars, football matches and other social gatherings in the villages as explained by some participants:

> Men have inherited the beliefs that they are superior to women. Women are voiceless in making and influencing decisions at the household. Even the young generation has inherited the same cultural beliefs. It is difficult for women to educate or communicate to men the information they receive from the health facilities. We need to identify male champions who will talk to their fellow men in their social gatherings (Religious leader).

Other participants added:

> Most of the maternal and child health programmes are directed to women. Men do not get opportunities to receive health information becuaes they rarely attend clinics. Unfortunately, when women get the knowledge, they cannot deliver to their partners.

**Table 7** Barriers to male involvement in maternal and child health

| Problem | Root causes | Strategies to address root causes |
|---|---|---|
| Low male involvement in maternal and child health | Lack of health education, especially for men. Most often health education is provided by health workers during ANC clinics. Men rarely attended ANC clinics | ▶ Male champions providing education to men during social gatherings<br>▶ Community and religious leaders providing education during community meetings |
| | Traditional gender roles and norms. Maternal and child health is considered as women's affair. Men's role is mainly supporting women financially | ▶ Male champions providing education to men<br>▶ Community and religious leaders providing education during community meetings |
| | Fear of HIV testing. As part of the PMTCT programme, couples are required to test for HIV during ANC appointment. Men always prefer their wives to test first and assume that the wives' results would be the same to the husband. | ▶ Male champions providing education to men on importance of couple HIV counselling and testing<br>▶ Community and religious leaders providing education during community meetings on importance of couple HIV counselling and testing |
| | Low birth spacing makes men fear to accompany their wives. Men think that they would be reprimanded by health providers and fellow men | ▶ Male champions providing education to men on the importance of family planning<br>▶ Sensitising healthcare to provide friendly services to couples attending clinics |
| | Couple relationships may encourage or hinder male participation in maternal and child health matters. The better the relationship, the higher the male participation and vice versa. | ▶ Women group members sensitising fellow women on the importance of better couple relationship |
| | Unfavourable environment in the health facilities. Most mentioned environments including long waiting time, disrespectful languages of the health providers, and lack of physical space to accommodate men and women concurrently. | ▶ Healthcare workers to provide friendly services to couples attending clinics<br>▶ The local government to improve physical space to accommodate men and women concurrently<br>▶ Engaging health facility governing committees to improve health facilities |

ANC, antenatal care; PMTCT, prevention of mother to child transmission.

They always fear to talk to their husbands (Female participant).

We request IMCHA project team to focus mostly on men so that we can get first-hand information instead of waiting for female partners. We have always been excluded when it comes to maternal and child health education. Women are given education in health facilities. Men rarely get these opportunities (Male participant).

Closely connected to the above, participants in various meetings underlined the importance of engaging male gatekeepers particularly religious and community leaders. It was highlighted that religious and community leaders are highly respected in the community. They have the power to convey health information to men and women alike. According to our participants, male gatekeepers should be actively engaged in the sensitisation campaigns or in providing education to men and women in their communities as explained by some respondents:

We should think very carefully about people who will deliver messages or education to the community. Others are an obstacle and using them as champions will end up losing instead of gaining. People will not listen; instead they will start whispering. But if we use influential people like religious leaders, elderly, or

government officials, we might be able to increase the acceptability of the messages (Community leader).

Another respondent added:

Women have presented the truth. The challenge is to get the right people who will deliver the key messages to the community and men in particular. I think if we work with religious and community leaders, the messages can easily be accepted by men (Male participant).

Almost all participants requested the IMCHA project team to provide training to male champions and gatekeepers. They also requested to provide them with key messages that should be delivered to the community, particularly to men. The key messages should focus on the importance of men to attend ANC clinics along with their partners, birth preparedness including shared decision making during pregnancy and child birth and the importance of FP use.

This sensitization and training should not stop today. Women have presented very important issues which need to be addressed in our communities. We request researchers to provide more training to men, religious, and community leaders so that they can deliver these messages to the community. We have

various platforms in our communities which can be used to communicate this information (Government leader).

## Women champions and education of fellow women

There was consensus among participants that relationships among couples may encourage or hinder male participation in MCH matters; on the one hand, unfriendly relationship will hinder active participation of men. On the other hand, good couple relationship will facilitate male involvement in MCH. In this context, the use of women groups was considered important, particularly to provide education to their fellow women on the best practices. It was reported that women have many social platforms where they could share and discuss the information related to couple relationships during pregnancy and child birth. Improved couple relationships will increase male participation in MCH as elaborated by some participants:

> Men are like children. When you need to talk to them, you should read their mind and feelings. If they are in bad mood don't say any thing to them, wait until they feel better. We need to understand the moods of our husbands and the best occasion that men can listen and understand our needs (Female participant).

Other participants added that:

> We, women, have problems. When a man agrees to do some household chores, we still want to make it public. We speak to our friends, neighbours or in saloons that, you just make my hair slowly, no need for hurry; my husband is taking care of children…. Even food, I will find it already cooked. If men know this, they will never support you again (Female participant).

> The way a wife communicates information to the husband may result in participation or non participation in antenatal care clinics. Some women think that when they are pregnant it is the time to harass their husbands. Men need soft language, if you use harsh or rude language, we interpret it as a command. Usually, men need respect and love. If you show us respect and love, we are ready to accompany you to ANC clinics and assist in domestic chores (Male participant).

Another participant had this to say:

> Women are the root cause of the low male involvement because they do not communicate with their husbands in a polite language. For example, when I come back home and tell my wife, I need the night meal (to have sex) and the wife replies rudely that 'I'm not ready', I will keep quiet. If I get the same answer the following day….I may start thinking of alternatives, looking for other women to satisfy my sexual desire. So, women need to use soft language, and we can understand them and plan together (Male participant).

## Discussion

This study adds new knowledge on the strategies for promoting male involvement in MCH in low-income and middle-income countries. It is one of the few to have used action research approach to promote male involvement inMCH. While action research has been used extensively in the health sector in Tanzania[30–32] and in other low-income and middle-income countries,[33–36] there has been little evidence of its appropriateness in promoting male involvement in MCH. We have demonstrated how stakeholders, including men, were able to design culturally acceptable strategies to increase male involvement in MCH. The next section discusses the strategies suggested by the participants in the current study in the light of other studies in low-income and middle-income countries.

### Health education and community sensitisation

Health education and community sensitisation campaigns were widely suggested in many villages as an important strategy which can be used to promote male participation in MCH. Participants felt that majority of community members, particularly men, lack MCH education. While male champions who participated in the study had adequate knowledge of ANC care, FP and EBF, this was only a small segment of the population. In addition, male champions were selected based on their interests in MCH matters. It may be possible that most of the male champions had experience in MCH matters. Health education interventions tailored to women and men have been effective in increasing knowledge, health seeking behaviour and raising awareness on issues related to MCH.[18 37–39] In our study, participants said that in order to make it sustainable, health education should be provided by healthcare workers and CHWs who already exist in the communities. They also preferred the campaigns to be integrated in the community and political gatherings in their respective villages. Participants preferred that key messages be developed to facilitate the campaigns and sensitisation programmes.

Connected to the above, participants suggested the use of male champions and male gatekeepers to promote behavioural change among men. It was evident from the findings that while women had access to health education during ANC clinics, men rarely attend health facilities. However, due to entrenched gender power relations, most often women are not able to convey the health information received from health facilities to their husbands. A study in Tanzania showed that the traditional way of using women to convey health information to male partners was a barrier to male involvement because men preferred to receive the message through other men.[40] The use of male champions is an effective strategy to include and engage men by targeting men focused key messages on the importance of participating in MCH. A study in the Philippines reported that using male peer educators improved participation of men in MCH matters.[41] Similarly, recent studies in Malawi underscored the importance of male champions in facilitating men's

participation in MCH and challenging traditional gender norms.[19 42] Men were targeted with messages focused on the benefits of utilising MCH services.[19 42] However, while in the Malawian studies male champions delivered the key messages to men secretly,[19] in our study, the participants suggested that male champions should discuss MCH openly in the men's platforms including football matches, bars and other places where men meet. Furthermore, sensitisation and support should also be provided to religious, traditional, and government leaders to spread these messages in various religious, political and social gatherings. Studies in other Africa countries have highlighted the importance of engaging religious and traditional leaders in promoting male involvement in MCH.[18 19 42]

Furthermore, participants underlined the importance of managing couple relationships in order to increase men's participation in MCH services. Female and male participants suggested that women should learn how to strategically manage their couple relationships in order not to deter male men from engaging in MCH. It is absolutely the case that men's ability and willingness to be involved in MCH is influenced by couple relationship dynamics.[19 42] However, in our study, it seems that participants recommended that the burden of emotional labour involved in managing couple relationships should fall exclusively to women, and not to men. For example, it is the women and not men who are responsible for communicating with 'soft language'. While this finding is an important contribution to the literature, it implies the dominance of unequal expectations of men and women in the society. Managing couple relationships should be the responsibility of both women and men.

### Use of health facility governing committees

In our study, participants suggested the use of health facility governing committees to address facility-related barriers to male involvement in MCH. Health facility governing committees have been reported to significantly facilitate access to MCH services.[43] However, in Tanzania, studies have indicated that while health facility committees exist in all public primary healthcare facilities, limited financial resources, limited capacity and skills, attitudes of health workers and lack of clarity of the role and mandate of committees affect the performance of the committees.[44–46] In order to maximise their performance, deliberate efforts should be made to strengthen the capacity of the health committees. Furthermore, the Ministry of Health should ensure that district Councils allocate adequate funds to the district annually to support meetings of the health committees as it is done for the council Health Services Boards (CHSBs). Studies have reported that CHSBs in Tanzania managed to attract members with good qualification mainly due to the availability of good incentives.[45 46]

### Limitations of the study

The main limitation of this study is that it does not report the implementation and effects of the designed strategies. The findings reported are based on the initial stages of the PAR. The programme is now going into the fourth and fifth phases of implementation and evaluation of the effects of the strategies on male involvement in MCH. Another limitation of the study is lack of independence of the data collection. The data collection was carried out by the researchers who were part of the ART. However, the data collectors had no motivation to consciously or subconsciously skew the responses in a particular direction, although errors in data collection will always be expected. The data collectors were paid to collect the data as accurately as possible and their compensation did not depend on what data they collected. Moreover, the interviewers did not collect data from their family or friends and had no personal stake in the collected data.

### Conclusion

While action research has been used extensively in the health sector, there has been little evidence of its relevance to promoting male involvement in MCH. This study is one of the few to have used action research approach to promote male involvement in MCH. We have demonstrated how stakeholders, including men, were able to design culturally acceptable strategies to increase male involvement in MCH. The use of participatory approach empowered communities, including women, to diagnose barriers to male involvement and develop culturally acceptable strategies to address these barriers. Our participatory approach to male involvement may likely increase sustainability of the interventions beyond the life span of the project. The IMCHA programme is now going into the last phase of implementation and evaluation of effects of these strategies on MCH outcomes. More lessons will be identified during the implementation of these strategies.

**Acknowledgements** This study was part of the Innovating for Maternal and Child Health in Africa (IMCHA) programme, which is being implemented in Kilolo and Mufindi districts in Iringa Region, Tanzania. We are grateful to the regional, district and local level stakeholders for participating in this study.

**Contributors** SM, PK, KB and MA conceptualised the study, including developing data collection tools. SM, PK and PJ collected, coded and analysed data. PJ and SM drafted the manuscript. PK, SF, KB and MA contributed to the manuscript writing. All authors approved the final manuscript.

**Funding** The study was funded by a joint initiative of three Canadian government agencies, namely; the Canadian Institutes of Health Research (CIHR), Global Affairs Canada and Canada's International Development Research Centre (IDRC) under IMCHA programme grant no: IDRC 108023-001.

**Competing interests** None declared.

**Patient consent for publication** Not required.

**Ethics approval** This study got approval from the Ethical Committee of the Medical Research Council of Tanzania (NIMR) No: NIMR/HQ/R.8a/Vol.IX/2119. The study was also approved by the Regional Administrative Officers in the region of study.

**Provenance and peer review** Not commissioned; externally peer reviewed.

**Data availability statement** The datasets are not publicly available since participants did not give consent for the public sharing of their information. However, summaries of the information and data collection tools are available from the corresponding author on request.

**ORCID iD**
Stephen Maluka http://orcid.org/0000-0002-0369-0858

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
