## [Reviewer comments · BMJ Open]

ARTICLE DETAILS

TITLE (PROVISIONAL)	Leaving no one behind: using action research to promote male involvement in maternal and child health in Iringa region, Tanzania
AUTHORS	Maluka, Stephen; Japhet, Paul; Fitzgerald, Sian; Begum, Khadija; Alexander, May; Kamuzora, Peter

VERSION 1 – REVIEW

REVIEWER	Ronee Wilson, PhD University of South Florida College of Public Health USA
REVIEW RETURNED	13-Apr-2020

GENERAL COMMENTS	The authors have detailed their participatory action research to promote male involvement in MCH in Tanzania. This is important work; however, there are several areas of this manuscript that need to be addressed before it can contribute meaningfully to the male involvement literature. Introduction: Page 4: Line 8 – 99% of MCH deaths or all deaths in general? Page 4: Second paragraph: Line 16 – Should be a period or full stop after recommended visits. The next sentence should start “Additionally, only 24% of women.....”. Lines 18-20 – Sentences should be restructured to reflect that 63% of deliveries in a health facility means 37% were home deliveries and 64% of deliveries assisted by health professional means 36% were not. Page 4: Line 46 – “use of by-laws” needs clarification. Page 4: Line 53 – After”mother to child transmission” add “of Human Immunodeficiency Virus (HIV)”. There is some redundancy in the language used in the Introduction. Lines 48/49 are very similar to 60. Methods: Figure 1: If there were Women and Men Groups why does Figure 1 only highlight the Women Groups Action Cycle for identifying barriers to male involvement? What did the Action Cycle look like for Men Groups? 2.1.1 Formation of Women and Men Groups need clarification. Lines 46 and 47 says women and men groups comprised 20 and 10 members. Lines 52 and 53 says 400 women and 200 men were identified and included in groups. Did these 400 women and 200 men make up several groups of 20 and 10, respectively. This needs clarification. Page 5: Line 55 – Who conducted the training? Was it standardized? Page 5: Line 57 – Were women and CHWs the only ones designing
--

	strategies to address barriers to the access and use of MCH services? Were men able to provide 2.1.2 Who were the male champions? Why weren't men groups assembled to identify barriers? Were the male champions the same men included in the male engagement strategy discussions? 2.1.3. Page 6. Line 22 – Please clarify “members of the user committees”. Table 1: In relation to the “20 intervention” villages mentioned in Section 2.1.1, what is Table 1 demonstrating? Are Wards 1 -10, 10 of the 20 villages? Table 2: What does S/N stand for? How does Table 2 align with 20 women groups and the 10 men groups mentioned in 2.1.1? What does “preparing women group members for the community meetings” entail and does it differ from “training of women group members”? 2.2 Were there five local interviewers in each village or did the same five interviewers cover all 20 villages. Mentions permission to audio-record but does not mention participant consent or IRB review. Findings: Table 3: Who are the women in Table 3? Why don't the numbers add up to the 400 women mentioned in 2.1.1? Where is Table 4? Table 5: Who are the male champions or men in Table 5? Do they meet the criteria for male participants (having a wife/female partner and experience with childbirth)? Table 6: Does n represent the number of men or the number of responses. None of the sections add up to 175. Table 7: Table 7 would be more informative if the authors added a third column that indicates the strategy that will be used to address the stated Root Cause. Qualitative findings are very interesting.
--	---

REVIEWER	Mohammed, Bedru Hussen The University of Hong Kong, Hong Kong
REVIEW RETURNED	15-Jun-2020

GENERAL COMMENTS	The authors clearly stated that: "In this paper, we report the first three phases of the PAR as the last two phases (implementation and evaluation) were not yet completed." At the same time, they declared that "Our participatory approach to male involvement increases the sustainability of the interventions beyond the life span of the project." It is not clear how they reach this conclusion before the project completion ie evaluation stage.
--

REVIEWER	Gina Clarkson Idaho State University, USA
REVIEW RETURNED	06-Jul-2020

GENERAL COMMENTS	Review First, I would like to say that this is a great study and needed research. I enjoyed reading this very much. I agree that little is known in this area. I think that you used appropriate methodology to begin to address this issue. Recommendations:
---

	1. I noticed that there was not a reporting guide noted in the references. There were some questions that I had about the methodology and I think this oversight could be corrected by using a proper reporting guide. I went to the EQUATOR website and found this reference which might be helpful for you: Smith L, Rosenzweig L, Schmidt M. Best practices in the reporting of participatory action research: Embracing both the forest and the trees. The Counselling Psychologist. 2010;38(8):1115-1138. 2. The introduction could use some comparison statistics so that your reader can compare Tanzania's maternal/child morbidity and mortality to that of another country. Your western readers might gain some insight if you use a western country's statistics for comparison. 3. What is the estimated morbidity and mortality from poor prenatal care and low rate of assisted deliveries in Tanzania? How is it estimated that father involvement would help with these statistics? 4. In the two districts (or at least the region...) where the study took place, what are the birth statistics? 5. How were your subjects recruited? 6. How were the officials invited? How many who were invited did not attend and why? 7. What did the training entail for the CHWS and the women's group members? 8. Regarding table 1, I don't understand the significance of why there are totals for each type of participant. I also don't understand the relevance of why each type of participant and numbers of those are listed in such detail. Please explain why it is important for your reader to know the types and numbers of officials in attendance at each meeting. Also, some definitions of what the different types of officials are and their duties might be helpful for someone not familiar with Tanzanian government to understand. What was the gender of the government officials? Do you think the gender of the official had any effect on their willingness to participate? 9. Was everyone at the meeting on board with creating change or did you have dissent where some didn't think it was a husband's place to participate in the birth process? 10. Criteria for the female group members was listed in the methodology as being ages 15 to 49, yet the results list age as up to 61. Please explain this discrepancy. 11. In many countries, people under the age of 18 are not considered as able to provide consent. What is the age of consent in Tanzania? 12. How did you conduct ethics and consent process for participation? 13. Please explain the relevance of figure 2. 14. How many in the study were actually pregnant at the time of the meetings? How important was it for the participants to understand the current state of male
--	---

	attendance at prenatal appointments and birth? With many older women, how would they be familiar with this process and wouldn't they be more likely to want to stick to the 'old' ways? 15. Why was it important to note if participants were farmers or not? 16. Why was it important to report the number of children that the men had but not the women? What was the significance of number of children and how might this have affected a participant's responses in the meetings? 17. Limitations should address more of the methodological limitations, for example, subject recruitment, data analysis, and a more thorough description of how many researchers participated in the thematic analyses and their qualifications. Please pay particular attention to any bias that might be found in your study. 18. In section 4.1, the last sentence says that it 'should' be the responsibility of both partners in a relationship, although I agree with you, I think instead of a judgement, perhaps provide some literature (research studies) on the importance of both male and female participation.
--	--

VERSION 1 – AUTHOR RESPONSE

Reviewers' Comments	Response to Comments
Reviewer: 1	
Page 4: Line 8 – 99% of MCH deaths or all deaths in general?	We have clarified that it is MCH deaths
Page 4: Second paragraph: Line 16 – Should be a period or full stop after recommended visits. The next sentence should start “Additionally, only 24% of women.....”.	Changes effected
Lines 18-20 – Sentences should be restructured to reflect that 63% of deliveries in a health facility means 37% were home deliveries and 64% of deliveries assisted by health professional means 36% were not.	Changes effected
Page 4: Line 53 – After”mother to child transmission” add “of Human Immunodeficiency Virus (HIV)”.	Changes effected
There is some redundancy in the language used in the Introduction. Lines 48/49 are very similar to 60.	We have deleted the sentence which was in lines 48/49
Figure 1: If there were Women and Men Groups why does Figure 1 only highlight the Women Groups Action Cycle for identifying barriers to male involvement? What did the Action Cycle look like for Men Groups?	We have clarified that men were involved in different stages of the Women group meetings. There was to different cycle of meetings for men.
2.1.1 Formation of Women and Men Groups need clarification. Lines 46 and 47 says women and men groups comprised 20 and 10 members. Lines 52 and 53 says 400 women and 200 men were identified and included in groups. Did these 400 women and 200 men make up several groups of 20 and 10, respectively.	We have clarified this paragraph
Page 5: Line 55 – Who conducted the training? Was it standardized?	Clarification has been provided. See pg. 5

Page 5: Line 57 – Were women and CHWs the only ones designing strategies to address barriers to the access and use of MCH services? Were men able to provide	We have clarified that men were involved in commenting on the barriers and jointly designing and implement strategies. See pg. 5
2.1.2 Who were the male champions? Why weren't men groups assembled to identify barriers? Were the male champions the same men included in the male engagement strategy discussions?	The male champions were the same men included in the male engagement strategy discussions.
2.1.3. Page 6. Line 22 – Please clarify “members of the user committees”.	We have clarified that these were members of the health facility governing committee which in Tanzania exist in each health facility (Dispensary, health centre ad Hospital)
Table 1: In relation to the “20 intervention” villages mentioned in Section 2.1.1, what is Table 1 demonstrating? Are Wards 1 -10, 10 of the 20 villages?	We have amended this Table
Table 2: What does S/N stand for?	We have deleted this as it adds no value
How does Table 2 align with 20 women groups and the 10 men groups mentioned in 2.1.1?	We have indicated that questionnaires and interviews were conducted with a sub-set of women groups and men (male champions); not all 400 women and 200 men who participated in the PAR cycle.
What does “preparing women group members for the community meetings” entail and does it differ from “training of women group members”?	We have clarified this
2.2 Were there five local interviewers in each village or did the same five interviewers cover all 20 villages.	We have clarified that the same five interviewers cover all 20 villages.
Mentions permission to audio-record but does not mention participant consent or IRB review.	IRB review process is indicated at the end of the manuscript as per requirements of the Journal.
Table 3: Who are the women in Table 3? Why don't the numbers add up to the 400 women mentioned in 2.1.1?	We have indicated that questionnaires and interviews were conducted with a sub-set of women groups; not all 400 women who participated in the PAR cycle.
Table 6: Does n represent the number of men or the number of responses.	We have indicated that questionnaires and interviews were conducted with a sub-set of men (male champions); not all 200 men who participated in the PAR cycle.
Table 7: Table 7 would be more informative if the authors added a third column that indicates the strategy that will be used to address the stated Root Cause.	We have added a column which indicate suggested strategies
Reviewer: 2	
The authors clearly stated that: "In this paper, we report the first three phases of the PAR as the last two phases (implementation and evaluation) were not yet completed." At the same time, they declared that "Our participatory approach to male involvement increases	We have revised this sentence

the sustainability of the interventions beyond the life span of the project." It is not clear how they reach this conclusion before the project completion ie evaluation stage.	
Reviewer: 3	
I noticed that there was not a reporting guide noted in the references. There were some questions that I had about the methodology and I think this oversight could be corrected by using a proper reporting guide.	We did not understand this comments. We have provided detailed description of our PAR.
The introduction could use some comparison statistics so that your reader can compare Tanzania's maternal/child morbidity and mortality to that of another country. Your western readers might gain some insight if you use a western country's statistics for comparison	We feel it difficult because there many countries in the World. We are not sure which Western country would compare best with Tanzania.
In the two districts (or at least the region...) where the study took place, what are the birth statistics?	We have added a Table in the Methodology which shows these statistics
How were your subjects recruited?	We have clarified this in the methodology section
How were the officials invited? How many who were invited did not attend and why?	We have that community leaders were used to recruit officials. We did not systematically keep data of those who did not attend.
What did the training entail for the CHWS and the women's group members?	We have clarified the content of the Training. We have also attached supplementary materials for detailed descriptions of all phases of women group meetings.
Regarding Table 1, I don't understand the significance of why there are totals for each type of participant. Also, some definitions of what the different types of officials are and their duties might be helpful for someone not familiar with Tanzanian government to understand.	We have revised this Table
Was everyone at the meeting on board with creating change or did you have dissent where some didn't think it was a husband's place to participate in the birth process?	Participants were purposively selected by community members. Only those who were considered by the community as key change agents were involved in the meetings. Was everyone at the meeting on board with creating change
Criteria for the female group members was listed in the methodology as being ages 15 to 49, yet the results list age as up to 61. Please explain this discrepancy.	We have clarified this discrepancy. See 1 st parag on pg. 5.
How did you conduct ethics and consent process for participation?	We have elaborated this. See pg. 7.
How many in the study were actually pregnant at the time of the meetings? How important was it for the participants to understand the current state of male attendance at prenatal	We did not track number of pregnant women in the groups.

appointments and birth?	
With many older women, how would they be familiar with this process and wouldn't they be more likely to want to stick to the 'old' ways?	We did not notice this in the PRA process.
Why was it important to report the number of children that the men had but not the women? What was the significance of number of children and how might this have affected a participant's responses in the meetings?	This was important in order to determine men's willingness to use family planning methods.
Limitations should address more of the methodological limitations, for example, subject recruitment, data analysis, and a more thorough description of how many researchers participated in the thematic analyses and their qualifications. Please pay particular attention to any bias that might be found in your study	We have elaborated these issues in the limitations including potential bias that might have been introduced.

VERSION 2 – REVIEW

REVIEWER	Ronee Wilson USF, USA
REVIEW RETURNED	12-Sep-2020
GENERAL COMMENTS	The revised version of the manuscript is much improved.